# Precisely translating computed tomography diagnosis accuracy into therapeutic intervention by a carbon-iodine conjugated polymer

Mingming Yin[1], Xiaoming Liu [2,3], Ziqiao Lei[2,3], Yuting Gao[4], Jiacheng Liu[2,3], Sidan Tian[1], Zhiwen Liang[5], Ye Wang[5], Fanling Meng[1,6,7] & Liang Luo [1,6,7✉]

X-ray computed tomography (CT) has an important role in precision medicine. However, CT contrast agents with high efficiency and the ability to translate diagnostic accuracy into therapeutic intervention are scarce. Here, poly(diiododiacetylene) (PIDA), a conjugated polymer composed of only carbon and iodine atoms, is reported as an efficient CT contrast agent to bridge CT diagnostic imaging with therapeutic intervention. PIDA has a high iodine payload (>84 wt%), and the aggregation of nanofibrous PIDA can further amplify CT intensity and has improved geometrical and positional stability in vivo. Moreover, with a conjugated backbone, PIDA is in deep blue color, making it dually visible by both CT imaging and the naked eyes. The performance of PIDA in CT-guided preoperative planning and visualization-guided surgery is validated using orthotopic xenograft rat models. In addition, PIDA excels clinical fiducial markers of imaging-guided radiotherapy in efficiency and bio-compatibility, and exhibits successful guidance of robotic radiotherapy on Beagles, demonstrating clinical potential to translate CT diagnosis accuracy into therapeutic intervention for precision medicine.

[1] National Engineering Research Center for Nanomedicine, College of Life Science and Technology, Huazhong University of Science and Technology, 430074 Wuhan, China. [2] Department of Radiology, Union Hospital, Tongji Medical College, Huazhong University of Science and Technology, 430022 Wuhan, China. [3] Hubei Province Key Laboratory of Molecular Imaging, 430022 Wuhan, China. [4] Faculty of Materials Science and Chemistry, China University of Geosciences, 430074 Wuhan, China. [5] Cancer Center, Union Hospital, Tongji Medical College, Huazhong University of Science and Technology, 430022 Wuhan, China. [6] Key Laboratory of Molecular Biophysics of the Ministry of Education, College of Life Science and Technology, Huazhong University of Science and Technology, 430074 Wuhan, China. [7] Hubei Key Laboratory of Bioinorganic Chemistry and Materia Medica, School of Chemistry and Chemical Engineering, Huazhong University of Science and Technology, 430074 Wuhan, China. ✉email: liangluo@hust.edu.cn

X-ray computed tomography (CT) is one of the most powerful diagnosis and therapy guidance techniques, with a leading frequency of use and hospital availability[1–4]. Contrast agents with high X-ray attenuation, which typically contain high atomic number elements such as iodine, barium, and gold, are required to increase CT sensitivity and to better visualize the tissue of interest[5–8], especially when identifying solid malignancies with small sizes and at early stages[9–11]. However, current clinically available CT contrast agents, majorly based on iodinated small compounds, suffer from rapid clearance and insufficient effectiveness—only large doses at molar concentrations or tens of grams can provide adequate CT imaging contrast[12–14]. In addition, while modern multimodal imaging techniques can identify tumor lesions at increasingly small sizes, the intraoperative localization and precise excision of these nonvisible and nonpalpable lesions become very difficult[15,16]. Isotope-based radio-guided localization[17–19] and colored/fluorescent dyes[20–25] have been developed to improve surgical guidance. However, these technologies are still afflicted with fast diffusion and short half-life of the marker materials. Moreover, most lesion positioning agents for instruments (e.g., CT imaging) cannot be readily visualized by surgeons in the operation procedures.

An ideal CT contrast agent that can bridge diagnostic imaging and therapeutic interventions should maximize the absolute CT attenuation difference between the tissue of interest and the surroundings by minimal doses, and maintain imaging properties throughout the course of preoperative CT scan and the designated therapies, which usually takes several hours to one day. More favorably, the contrast agent should be visible across CT imaging and naked eyes to improve the outcomes of surgical guidance. On the other hand, such contrast agents are highly desired in image-guided radiation therapy (IGRT), an emerging technology that has been introduced to improve the precision and lower the treatment-associated toxicities of radiotherapy[26–28]. IGRT essentially relies on the accurate positioning of lesions throughout the radiotherapy[29,30]. However, tumors rarely show a fixed location during irradiation or treatment due to respiratory movement, changes in organ filling, etc[31–33]. Radiopaque fiducial markers were typically implanted into the tumor tissue for tumor tracking and radiation beam alignment. Au-based fiducial markers are most commonly used in clinic, but they often lead to severe side effects, including inflammation, swollen, bleeding, and consequently position shift[34–36]. Their permanent in vivo stay is also a threat in prognosis and management. In addition, Au-based fiducial markers generate streak artifacts in CT images[37,38], therefore hindering the visibility of surrounding tissues and affecting the accuracy of dose distribution calculation and treatment planning[39,40]. CT contrast agents that can overcome these deficiencies with excellent biocompatibility and maximized performance are in urgent need to advance the technology of IGRT.

In this work, we demonstrate that poly(diiododiacetylene), or PIDA (Fig. 1a), represents a promising solution to the general problems of CT contrast agents, attributed to its high payload of iodine and many other intriguing characteristics. Polydiacetylenes have recently emerged as interesting, biocompatible functional materials for biomedical applications[41–44]. We and others have demonstrated the synthesis of a variety of functionalized polydiacetylenes through a co-crystal strategy[45–48]. PIDA contains a polydiacetylene backbone with only iodine atom substituents, and its iodine mass content can reach as high as 84.1%. Moreover, given its nanofibrous nature, PIDA can easily aggregate and accumulate on the target tissue for prolonged retention after being locally injected. In addition, the highly planar conjugated backbone of PIDA endows the material a deep blue color, so that it is easily distinguishable from the surrounding tissues by naked eyes, making it ideal for CT-guided preoperative planning and visualization-guided intraoperative target positioning. We have explored in this study the suitability of applying PIDA as a contrast agent for CT imaging, surgical guidance, and IGRT, in view of its ultrahigh X-ray attenuation efficiency, dual visibility by CT and naked eyes, and in vivo retention.

## Results

**Preparation of PIDA and ultrahigh X-ray attenuation.** The formation of PIDA requires the topochemical polymerization, i.e., the head-to-tail 1,4-polymerization, of diiodobutadiyne monomers assembled in an appropriate arrangement (Fig. 1a). To achieve the required spacial alignment of monomers for this topochemical polymerization, we use supramolecular scaffolds based on halogen bonds, the noncovalent interaction between iodoalkyne group in the monomer and functionalized Lewis bases[47,49–51]. In previous attempts to prepare PIDA[45,46,52], biscyanoalkyl or bispyridyl oxalamides were employed to align monomer guests in co-crystals. However, they could not yield PIDA with refined structure in large quantities[46], or only generated co-crystals that required high pressure to initiate the polymerization[53]. Here, using $N^1,N^2$-bis(2-(pyridin-3-yl)ethyl) oxalamide (**1**, structure shown in Supplementary Fig. 1) as the scaffold[54], we are able to produce PIDA co-crystals with a well-defined structure (Fig. 1b) on gram-scale, so that the possibility of further application of PIDA is boosted. PIDA•**1** co-crystals exhibit a metallic appearance that is characteristic of a high degree of polymerization (Fig. 1b, bottom right), and scanning electron microscope (SEM) image reveals their very smooth surface, with one dimension (010) much smaller than the other two dimensions (Fig. 1b, top right).

In order to isolate PIDA from the co-crystals and formulate it in aqueous media for in vivo applications, we utilized an amphiphilic polymer, polyethylene glycol-grafted poly(maleic anhydride-alt-1-octadecene), or C18-PMH-PEG, as the surfactant[55]. Sonication followed by dialysis of a mixture of the PIDA•**1** co-crystals and C18-PMH-PEG in water resulted in a stable blue aqueous suspension (Fig. 1c). The UV-visible absorption spectrum of the blue PIDA suspension (Fig. 1d) exhibited a maximal absorption peak at 652 nm with a broad shoulder at 750 nm, attributed to the planar backbone of the material. The Raman spectrum of the suspension was consistent with that of the co-crystals (Fig. 1e), indicating that the structure of PIDA was preserved during the process. Transmission electron microscope (TEM) imaging confirmed that the blue suspension mainly contained well-dispersed nanofibers with diameters of 30–50 nm (Fig. 1f). Moreover, the elemental analysis of the nanofibers by an energy-dispersive X-ray (EDX) detector showed the existence of only carbon (15.7 wt%) and iodine (84.3 wt%) atoms, which was consistent with the theoretical value of PIDA (15.9 wt% for carbon and 84.1 wt% for iodine) and evidenced the thorough removal of scaffold **1** (Fig. 1g).

The effective payload of iodine atoms is crucial to the high performance of CT contrast agents[12]. However, most of the current clinically available CT contrast agents have iodine mass contents of less than 50%, such as iohexol (46.4 wt%), iopromide (48.1 wt%), and iodixanol (49.1 wt%). The 84.1 wt% iodine content of PIDA warranted an ultrahigh X-ray attenuation ability. The CT value, an indicator in terms of Hounsfield unit (HU) that measures the ability of a material to attenuate X-rays, of the PIDA suspension increased linearly with the iodine concentration (Fig. 1h), indicating that PIDA was well dispersed in the media. The slope of HU values against sample concentrations for PIDA was 1.8 times higher than that for iohexol, suggesting a superior X-ray attenuation ability of PIDA. More strikingly, when a PIDA suspension was condensed to form PIDA

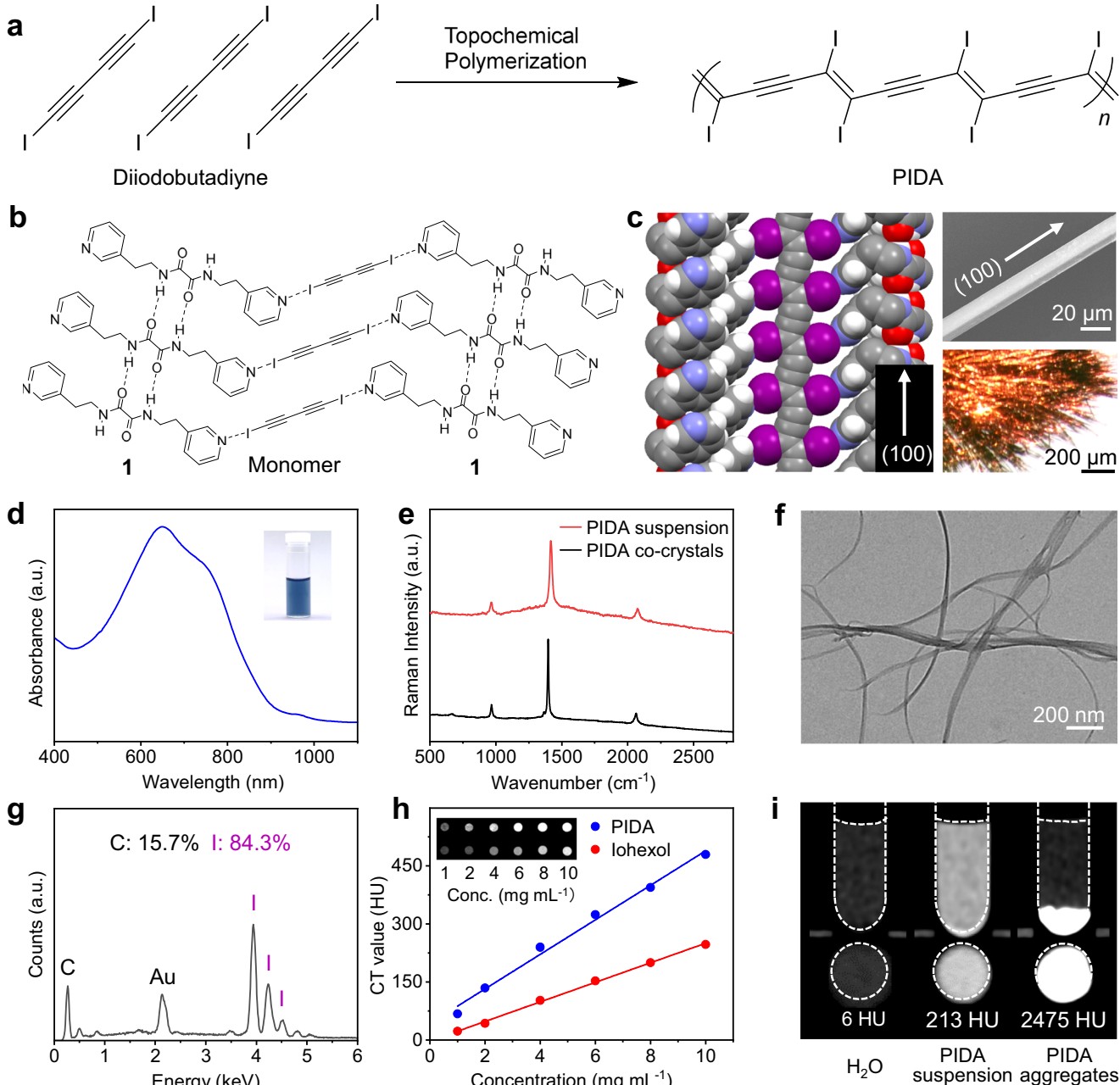

**Fig. 1 Preparation and characterization of PIDA. a** Synthesis of PIDA by topochemical polymerization. **b** Supramolecular scaffold formed between scaffold **1** and monomer diiodobutadiyne. **c** Single-crystal structure of PIDA●**1** co-crystals determined by X-ray diffraction (left), and SEM (top right) and optical (bottom right) images of PIDA●**1** co-crystals, with the long dimension corresponding to the (100) axis. Each experiment was repeated 3 times independently with similar results. **d** UV-vis absorption spectrum of the PIDA suspension. Inset: Picture of the PIDA suspension. **e** Raman spectra of PIDA●**1** co-crystals and the prepared PIDA suspension. **f** TEM image of dispersed PIDA nanofibers in the PIDA suspension. The experiment was repeated 3 times independently with similar results. **g** Elemental analysis of PIDA by EDX. **h** CT values of PIDA and iohexol as a function of sample concentrations. Inset: CT images of PIDA suspension and iohexol at corresponding concentrations. **i** CT images of pure water, PIDA suspension (5 mg mL$^{-1}$), and PIDA aggregates. Top row: side view; Bottom row: Top view. Source data are provided as a Source Data file.

aggregates (Supplementary Fig. 2), its CT value dramatically increased over 10 times, from 213 HU to 2475 HU (Fig. 1i). To validate this aggregation-induced CT intensity amplification, we injected a PIDA suspension and an iohexol solution (both with an iodine concentration of 4 mg mL$^{-1}$) into two porcine tissues, respectively (Supplementary Fig. 3). The contrast ratio of PIDA in the pork muscle tissue (87.8%) was 4 times higher than that of iohexol (21.7%), whereas in the pork fat tissue, the contrast ratio of PIDA (251.7%) was 17 times higher than that of iohexol (14.5%). In addition, PIDA nanofibers were observed to

accumulate in the injection sites, with a significantly longer in situ retention time than iohexol.

**PIDA-based ultraefficient CT imaging in rats.** Inspired by the remarkable X-ray attenuation ability of PIDA, we injected PIDA suspensions ([I]: 4 mg mL$^{-1}$) into rat legs to evaluate their performance in CT imaging. Iohexol solutions ([I]: 4 mg mL$^{-1}$) were injected into the control rat. Ideally, a contrast agent should improve the absolute CT attenuation of the target tissue to more than twice of the surrounding tissue and fluids[12]. The background

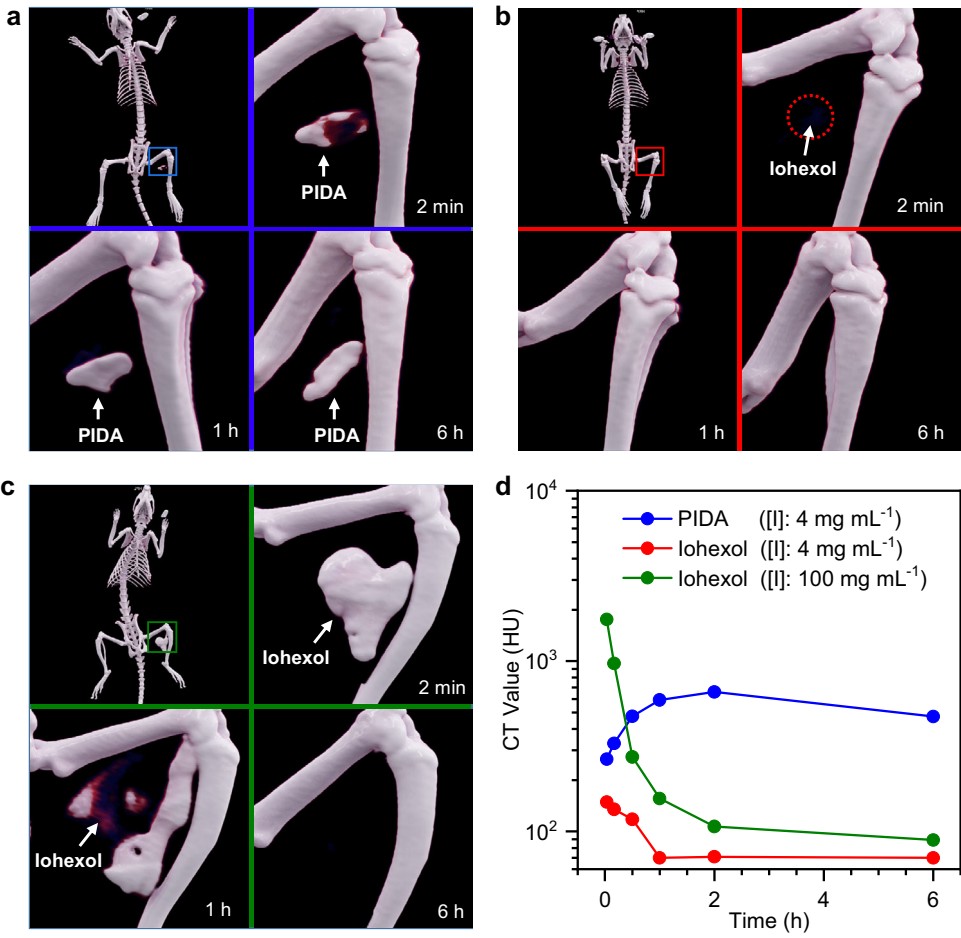

**Fig. 2 The performance of PIDA and iohexol in CT imaging of rats. a** CT images of a rat at different time after being injected with 200 μL PIDA suspension ([I]: 4 mg mL⁻¹). **b** CT images of a rat at different time after being injected with 200 μL of iohexol ([I]: 4 mg mL⁻¹). **c** CT images of a rat at different time after being injected with 200 μL of iohexol ([I]: 100 mg mL⁻¹). **d** Plots of CT values of different markers in (**a–c**) as a function of time post-injection. Source data are provided as a Source Data file.

CT signal of rat muscle tissue was about 50 HU, and the CT value of intramuscularly injected PIDA could reach 260 HU at 2 min after injection (Fig. 2a). As a comparison, the CT value of iohexol was around 150 HU at that time (Fig. 2b). Markedly, the CT value of PIDA continued to increase as time progressed and reached 660 HU at 2 h after injection, possibly attributed to the aggregation of PIDA at the injection site. The PIDA converged at the injection site and formed a well-defined marker with fixed geometry and position. Its CT value maintained at a high level of above 500 HU for over 6 h (Fig. 2d and Supplementary Fig. 4), a typical period required for CT-guided preoperative planning, surgery scheduling, and surgical procedure in hospital. As a comparison, the injected iohexol was barely visible at 2 min after injection, and its CT value dropped promptly to less than 100 HU within 1 h (Fig. 2b, d and Supplementary Fig. 5). Since an effective diagnostic dose of iohexol was typically in molar concentrations, we next applied a high concentration iohexol solution ([I]: 100 mg mL⁻¹) in rats. Although the CT value of the formed marker was very high initially (1700 HU), it decayed very fast and decreased dramatically down to 100 HU at 2 h post injection (Fig. 2c, d and Supplementary Fig. 6), suggesting the rapid diffusion and clearance of iohexol. The PIDA marker was also clearly visible under regular X-ray examination, which further confirmed the ultraefficient X-ray attenuation of PIDA (Supplementary Fig. 7). The great performance of PIDA in rat CT imaging hence demonstrates its great potential to meet clinical

marker needs, given its strong CT intensity at ultralow iodine concentrations, prolonged tissue retention time, as well as high geometrical and positional stability.

**CT and naked eyes dual-visible surgical marker.** The practicability of current imaging markers for surgical guidance is seriously restricted in that they cannot accurately translate diagnostic image findings into specific therapeutic intervention. A surgical marker that can bridge this gap by identifying occult lesions both under instrumental guidance and by direct visual observation is highly desirable in clinic[20,56]. In addition to the ultraefficient CT imaging characteristic, PIDA can preserve its unique deep blue color in vivo for over 1 week after the injection (Supplementary Fig. 8), hence enabling precise localization of the targeting lesions by both CT imaging and naked eyes. To evaluate if PIDA can serve as such a dual-visible surgical marker, we injected PIDA suspensions on the tumor periphery of an orthotopic xenograft rat model under CT imaging guidance (Fig. 3a), and examined the assistance of PIDA labeling in identifying tumor resection margin (Fig. 3b). As expected, the injected PIDA markers maintained high-level CT values with fixed geometry and positions on the tumor periphery for a period of 6 h (Fig. 3c and Supplementary Fig. 9). When we dissected the rats at 24 h after injection, the blue colored PIDA markers were still clearly visible by naked eyes and distributed around the nonpalpable tumor lesion to indicate the surgical

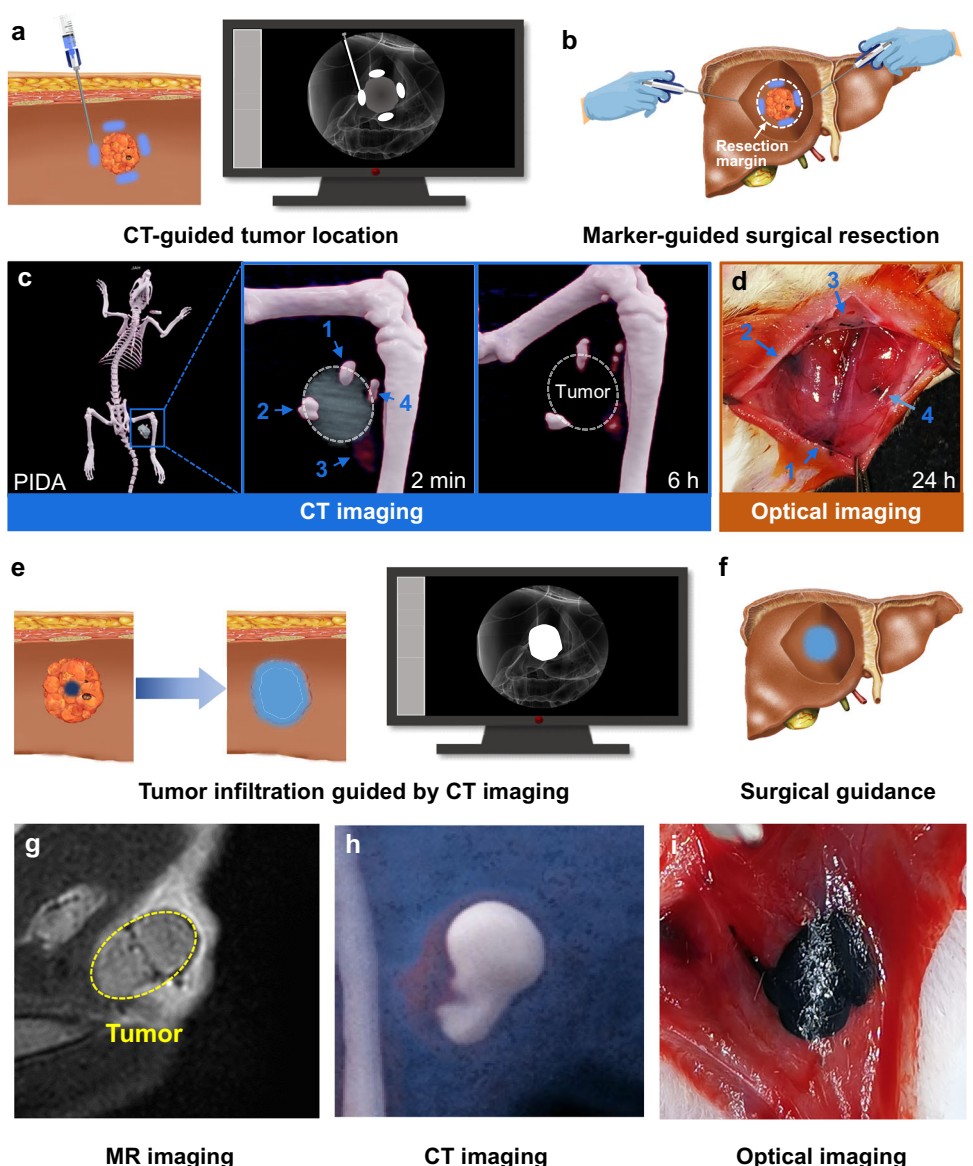

**Fig. 3 CT-guided preoperative tumor labeling and visualization-guided surgery on basis of CT and eye dually visible PIDA. a**, **b** Schematic illustration of tumor labeling by PIDA marker under CT guidance (**a**) and surgical resection of the labeled tumor guided by eye-visible blue PIDA (**b**). **c** CT images of a tumor-bearing rat injected with a PIDA suspension at 4 spots on the tumor periphery ([I]: 4 mg mL$^{-1}$, 50 μL each) over time. Tumors were omitted for better illustration for CT images of 6 h. **d** Exposed orthotopic tumor sites of the rat injected with PIDA. 4 blue PIDA residues surrounding the tumor clearly determined the resection margin of the tumor. **e** Schematic illustration of CT-guided tumor infiltration of intratumorally injected PIDA suspension. **f** Tumor infiltration of PIDA nanofibers for tumor boundary determination. **g**–**i** Complete tumor infiltration of PIDA nanofibers at 2 h post the injection measured by MR imaging (**g**), CT imaging (**h**), and optical imaging (**i**).

margin of the targeted tumor tissue (Fig. 3d). As a comparison, the CT signals of iohexol control (at the same iodine dose) disappeared quickly (Supplementary Fig. 10). More strikingly, after we intratumorally injected the PIDA suspension, the PIDA nanofibers diffused and fulfilled the whole tumor tissue in 2 h (Fig. 3e), clearly demonstrating the margin line between the tumor tissue and the normal tissue. The complete tumor infiltration of PIDA nanofibers had been confirmed by magnetic resonance (MR) imaging (Fig. 3g) and CT imaging (Fig. 3h), which could be directly used to identify the tumor boundary by naked eyes in the surgery (Fig. 3f, i). Intraoperative identification of tumor boundary remains a great challenge in clinic, and it is especially difficult for small and nonpalpable tumor lesions. The dual-visible PIDA allowed precise intraoperative localization of occult orthotopic tumors, demonstrating the potential to

directly translate diagnosis accuracy to therapeutic intervention for enhanced clinical surgical outcomes.

**Fiducial marker for stereotactic body radiation therapy.** In addition to the surgical marker, PIDA can also serve as an excellent candidate for fiducial markers in stereotactic body radiation therapy (SBRT). SBRT is an advanced approach of IGRT to precision radiotherapy, which employs robotic radiotherapy (RRT) to deliver precise doses of radiation with extreme accuracy (Fig. 4a)[57,58]. Because of its high precision, SBRT has to take real-time tumor movement into account, while tumor motions during the respiratory cycle are a major contributor to targeting uncertainty. RRT utilizes an orthogonal X-ray imaging system and an optical respiratory tracking system to build a real-time correlation model between the tumor and the external

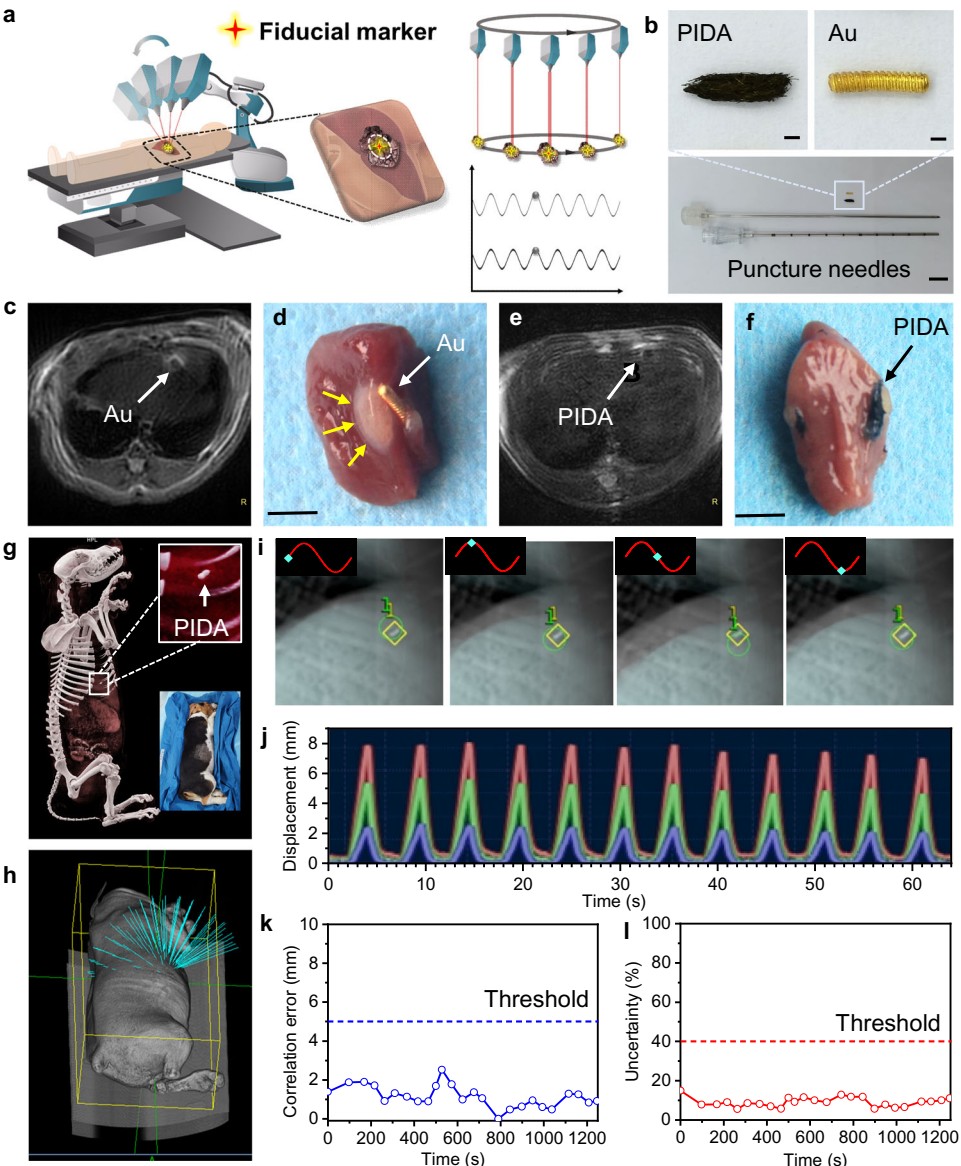

**Fig. 4 The evaluation of PIDA as a fiducial marker in robotic SBRT. a** Schematic diagram of robotic SBRT in a clinical setting. **b** Pictures of the puncture needles (Scale bars: 1 cm), the PIDA marker (Scale bars: 1 mm), and the Au marker used in clinic (Scale bars: 1 mm). **c** MR image of a rat implanted with an Au marker on Day 7 post the implantation. **d** Optical image of the resected rat liver tissue containing the Au marker, in which severe edema was observed (yellow arrows). Scale bar: 0.5 cm. **e** MR image of a rat implanted with a PIDA marker on Day 7 post the implantation. **f** Optical image of the resected rat liver tissue containing the PIDA marker. Scale bar: 0.5 cm. **g** CT image of the Beagle implanted with a PIDA marker. White rectangle: PIDA in the Beagle liver; Bottom inset: The Beagle fixed in a memory cushion for SBRT. **h** Three-dimensional distribution map of planned X-ray radiation for RRT on the Beagle. **i** X-ray images of the Beagle liver at different phases during the respiratory cycle. Green circles: the positions of the PIDA marker on live X-ray images; Yellow diamonds: the original position of the PIDA marker on the X-ray image simulated from preoperative CT modeling. **j** Beagle's respiratory movement during RRT determined by an optical respiratory tracking system. **k** Plot of the real-time correlation error during RRT. Blue dashed line indicates the threshold limit in clinic. **l** Plot of the real-time uncertainty during RRT. Red dashed line indicates the threshold limit in clinic. Source data are provided as a Source Data file.

abdominal surface of the patient. The established correlation model then allows the RRT to estimate tumor position based on optical tracking, which is sent to the robotic positioning system to enable real-time adjustment of the tumor position during respiratory cycles. In clinic, Au-based fiducial markers are most commonly implanted in tumor tissues to serve as a surrogate for tumor positioning in clinical practice. However, they are prone to cause local edema, inflammation, and positioning shifts[34–36,57], therefore impeding patients from receiving desired clinical benefits. PIDA, on the other hand, represents a straightforward solution to the problems associated with Au-based fiducial markers.

To make a direct comparison with the commercially available Au marker, PIDA was casted into a similar shape and implanted into the liver of a rat through puncture needles under the guidance of CT imaging (Fig. 4b). On Day 7 post the implantation, MR imaging revealed severe local edema caused by the implanted Au marker (Fig. 4c), which was confirmed by the optical image of the dissected tissue (yellow arrows, Fig. 4d). As a comparison, no PIDA-caused edema was observed either on the MR image (Fig. 4e) or dissected tissue (Fig. 4f). The blood analysis (Supplementary Table 1), the hematoxylin and eosin (H&E), and TdT-mediated dUTP nick end labeling (TUNEL) stain of the tissue around the implanted markers (Supplementary Fig. 11) also indicated excellent biocompatibility of

PIDA as the implantable marker. Moreover, the implanted Au marker contacted loosely to the surrounding tissues (Fig. 4d), which implied a high probability of position shift or even off target during clinical application. As a comparison, PIDA marker attached tightly to the surrounding tissues (Fig. 4f), suggesting a stable positioning in SBRT.

To further assess the positioning accuracy of PIDA as a fiducial marker in SBRT, we implanted a PIDA marker into the liver of a Beagle, on which a clinical SBRT was executed afterwards (Fig. 4g). A vacuum cushion was used to fix the posture of the Beagle during the SBRT treatment and to ensure the consistency with the preoperative CT modeling (Fig. 4g, bottom inset). From CT imaging, the PIDA marker exhibited a strong CT signal in the liver of the Beagle (Fig. 4g, top inset), on basis of which a three-dimensional distribution map of planned X-ray radiation was generated for RRT on the Beagle (Fig. 4h). The real-time position of the PIDA marker in RRT was monitored by live X-ray imaging (green circle in Fig. 4i), which moved close to the original position (yellow diamond in Fig. 4i, determined from the preoperative CT imaging) periodically during the respiratory cycle of the Beagle. In addition, the correlation error for RRT was measured based on the difference between the actual position of the target (green circle in Fig. 4i) and the predicted position of the target computed from the Beagle's respiratory movement (Fig. 4j). The average correlation error of the PIDA marker was $1.07 \pm 0.55$ mm (Fig. 4k), lower than that of Au markers typically found in clinic $(1.7 \pm 1.1$ mm$)$[59], indicating a good agreement with the correlation models for the execution of the planned RRT (Fig. 4h). The uncertainty of the PIDA marker, which provided the detection uncertainty value for the fiducial extraction algorithm, was $9.10 \pm 2.30\%$ (Fig. 4l), far below the uncertainty threshold parameter of 40%[60]. PIDA hence exhibited an exceptional positioning accuracy in SBRT of the Beagle.

An effective SBRT replies not only on the accurate positioning of fiducial markers in RRT, but also on the precise radiation dose calculation, which can be easily affected by artifacts in CT images. Compared with the nontrivial metal artifacts that were seen in the CT images of the rat implanted with an Au fiducial marker (Fig. 5a), PIDA presented a much more efficient labeling effect with minimal artifact interference (Fig. 5b). To further quantify the advantage of PIDA fiducial marker in artifact reduction, we used a thorax phantom with movable plugs to simulate the CT imaging of a whole human body in SBRT (Supplementary Fig. 12). Each movable plug in the phantom simulated a specific human organ, and inserting PIDA or Au marker to a plug mimicked the labeling of the corresponding organ. The CT imaging showed that PIDA could mark the target organ much more effectively than Au, given its negligible artifacts (Fig. 5c). The quantified CT intensities on the circled spots distributing near the PIDA marker (Fig. 5d, e) were almost identical to those of the unmarked sample (Control), suggesting that the artifact-free PIDA marker showed no influence on the visibility of the surrounding tissues. As a comparison, the Au marker exhibited a significant decrease of CT intensity in the tangential direction of the plug, as well as a considerable increase of CT intensity in the perpendicular normal direction. In addition, the calculated radiation dose distribution for RRT planning based on the Au marker apparently deviated from the actual dose distribution in the Control group (Fig. 5f, g). As a comparison, the PIDA marker resulted in a more accurate dose distribution calculation (Fig. 5g), showing a significant advantage over Au-based fiducial marker in assisting the treatment planning system for the precise radiation delivery in SBRT.

**Biocompatibility and Biodegradability of PIDA.** To further ensure the safety of PIDA for future applications as a CT contrast agent, we comprehensively evaluated its biocompatibility and biodegradability. We first mixed different concentrations of PIDA suspensions with rat red blood cells (RBCs). Triton was used as a positive control and set as 100% hemolysis. The hemolysis rates of all PIDA suspensions were below 5% (Fig. 6a), suggesting that PIDA did not cause hemolysis at tested concentrations. In addition, we incubated PIDA suspensions at different concentrations with 4T1 cells (mouse breast cancer cells), NIH 3T3 cells (mouse embryonic fibroblasts), and HEK 293T (human embryonic kidney cells) for 12 h (Fig. 6b). The cell viabilities of all PIDA-containing groups, tested by MTT assays, maintained above 90%, showing the good biocompatibility of PIDA.

Moreover, the weight of the experimental rats increased normally over time (Fig. 6c), and no significant difference in blood analysis and liver/kidney function tests between PIDA-injected rats and normal rats was observed (Supplementary Fig. 13), indicating that PIDA did not affect the physiological status of the animals. In addition, we have shown that the CT value of injected PIDA suspension ([I]: $4$ mg mL$^{-1}$) could maintain over 500 HU for 6 h (Fig. 2d). In fact, it still remained high at around 400 HU for 24 h (Fig. 6d, f), which should guarantee the completion of most CT-involved operations in hospital. The injected PIDA became invisible by CT imaging on Day 7 post injection (Fig. 6e), and on Day 21, the CT value decreased to a level close to iohexol ([I]: $4$ mg mL$^{-1}$), which was injected into another rat synchronously with PIDA (Fig. 6f). The CT value and the volume of the PIDA fiducial marker in rat liver also decreased over time (Supplementary Fig. 14). The gradual degradation and bioabsorbability of PIDA can be explained by the fact that polydiacetylene backbone is cleavable by reactive oxygen species[61], and the relatively labile carbon-iodine bond can also facilitate the process[62]. In addition, no obvious damage was found in major organs at 1 day or 1 week after the injection (Supplementary Fig. 15), and the H&E stains of main organ tissues also confirmed that PIDA did not cause any toxicity during the whole process (Fig. 6g). The tissue at the injection site exhibited some needle-caused damages, which completely healed after one month (Supplementary Fig. 16), further evidencing the nontoxic nature of PIDA. The excellent biosafety and biodegradability of PIDA therefore pave the way for translational studies of PIDA-based CT contrast agents.

## Discussion

The tremendous development of CT imaging has revolutionized its ability to accurately diagnose malignancies as well as to promote surgical interventions for improved clinical benefits. However, current clinically available contrast agents are less than optimal. Most iodinated contrast agents have rapid renal clearance and insufficient effectiveness, while metal-based contrast agents are hampered by strong artifacts and biosafety issues. In this work, we demonstrate that PIDA, a conjugated polymer of carbon and iodine, can address these deficiencies and serve as a contrast agent for CT imaging and related therapy intervention. With a high iodine payload of 84.1%, the intrinsic X-ray attenuation ability of PIDA is greater than most iodine-based contrast agents[12]. Moreover, PIDA exhibits aggregation-induced amplification in CT intensity attributed to its nanofibrous nature, leading to an over 10-fold boost in local CT intensity and significantly enhanced tissue retention as well as geometrical and positional stability. In addition, PIDA has a deep blue color because of its highly conjugated backbone, so that it can be dually visible by both CT imaging and naked eyes, making it a perfect marker candidate for CT-guided preoperative planning and intraoperative localization of nonpalpable tumor lesions. More notably, compared with metal-based fiducial markers, PIDA excels on its biocompatibility, fixed positioning, and

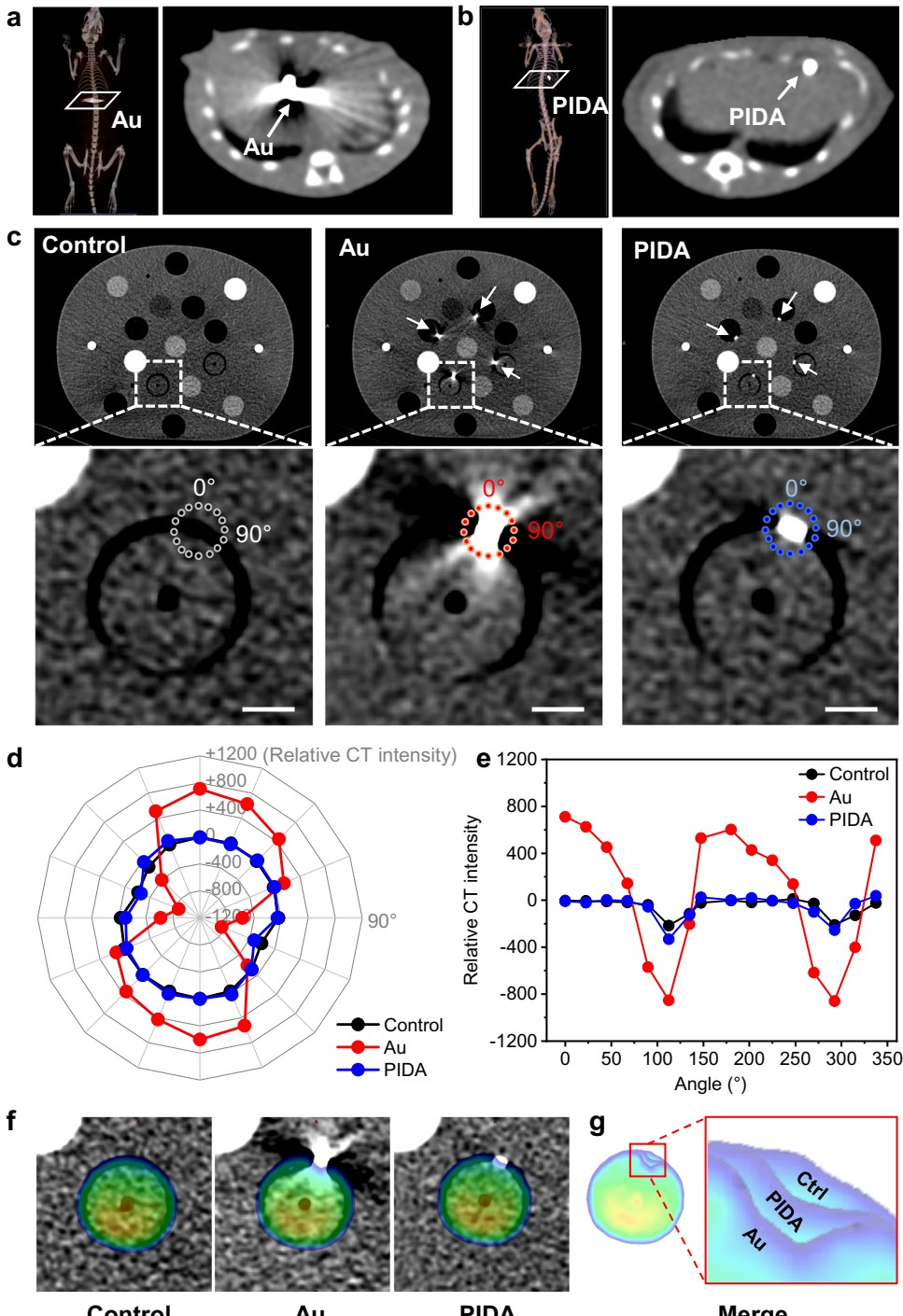

**Fig. 5 Accurate radiotherapy dose distribution planning based on PIDA fiducial marker. a** CT images of the rat implanted with an Au fiducial marker. **b** CT images of the rat implanted with a PIDA fiducial marker. **c** CT images of thorax phantoms with movable plugs. Scale bar: 1 cm. Control: Thorax phantom with no marker inserted; Au: Thorax phantom with 4 Au markers inserted (pointed by white arrows); PIDA: Thorax phantom with 4 PIDA markers inserted (pointed by white arrows). **d**, **e** Quantified relative CT intensities of circled spot arrays in (**c**). **f** Radiation dose distribution for RRT planning calculated on basis of the CT values of the corresponding thorax phantoms in **c**. Source data are provided as a Source Data file. **g** Overlapping of the calculated radiation dose distribution of Control (Ctrl), Au, and PIDA.

minimal artifacts while maintaining an effectively high CT intensity. These characteristics are critical for the precise calculation of dose distribution in treatment planning systems, as well as for the accurate radiation delivery in robotic SBRT.

Biomedical applications and clinical translations of conjugated molecular materials have been long pursued. The successful demonstration of the above advantages of PIDA on rat and Beagle models suggests it may have translational potential for clinical applications, not only expanding anatomical localization labeling which current technologies may be difficult to apply, but also encouraging the explorations of conjugated molecular materials to solve challenging clinical problems.

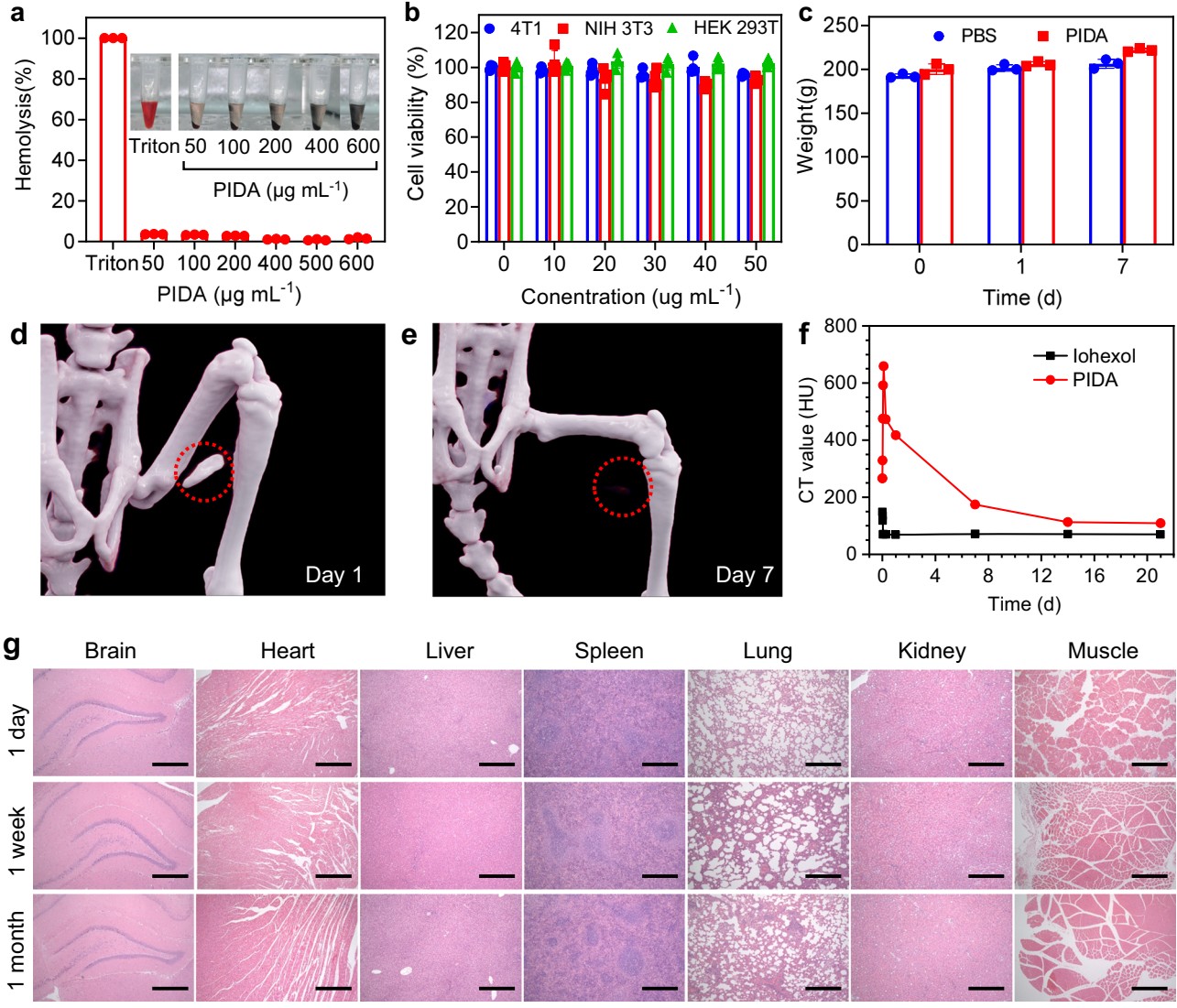

**Fig. 6 Biosafety assessment of PIDA. a** Hemolysis assay and images (insert) of RBCs incubated with different samples ($n = 3$ biologically independent samples). The percentage of RBC lysis was <5% for PIDA. **b** Viability of 4T1, NIH 3T3, and HEK 293 T cells pretreated with a series of doses of PIDA ($n = 4$ biologically independent samples). **c**. Body weights of rats received the treatment and control rats ($n = 3$ biologically independent samples). **d**, **e** CT images of PIDA in muscle at different time after injection. **f** Plots of CT values of PIDA and iohexol ([I]: 4 mg mL$^{-1}$, 200 μL) injected in rats as a function time. **g** H&E staining of main organs after being injected with PIDA. The experiment was repeated 3 times independently with similar results. Scale bars: 0.5 mm. All data are presented as Mean ± SD. Source data are provided as a Source Data file.

## Methods

**Synthesis of monomer diiodobutadiyne**. The monomer diiodobutadiyne (IDA) was synthesized through the reaction between 1,4-bis(trimethylsilyl)-1,3-butadiyne and N-iodosuccinimide catalyzed by silver nitrate[46]. 1,4-Bis(trimethylsilyl)-1,3-butadiyne (194.6 mg, 1 mmol) was dissolved in 30 mL of acetone. Silver nitrate (130.5 mg, 0.75 mmol) and N-iodosuccinimide (550.6 mg, 2.37 mmol) were added to the solution. The reaction flask was wrapped by aluminum foil to avoid the interference of ambient light. The reaction was stirred at room temperature for 4.5 h. (The completion of the reaction was confirmed by thin layer chromatography). After the reaction, the sample was extracted with n-hexane/water. The extraction process was repeated three times. The upper organic phase was dried with anhydrous sodium sulfate and concentrated in vacuo under ice bath. A light-yellow solid product was obtained (280 mg, yield 93%). $^{13}$C NMR (100 MHz, CDCl$_3$) δ (ppm): 79.86, −3.18[46].

**Synthesis of 1**. 2-(Pyridin-3-yl)ethan-1-amine (2.566 g, 21 mmol) was dissolved in 10 mL of acetone, and diethyl oxalate (1.461 g, 10 mmol) was added dropwise under stirring at room temperature[54]. After 6 h of reaction, the solvent was removed under vacuum. 40 mL of methanol was added to the remaining solid and heated to 50 °C to form a clear solution. The crystals gradually precipitated out of the solution at room temperature in 24 h. After removing the excess solution by filtration, the remaining solid was washed with cold methanol to obtain white

needle-like crystals (1.120 g, yield 45%).$^1$H NMR (400 MHz, CDCl3) δ 2.86 (t, 4H), 3.59 (q, 4H), 7.12(d, 2H), 7.50(s, 2H), 8.53(t, 2H), 8.54(t, 2H)[54].

**Preparation of PIDA•1 co-crystals**. Monomer diiodobutadiyne and **1** were dissolved in methanol at a molar ratio of 1:1. The solution was centrifuged at 640 × g for 10 min to remove possible dust impurities, followed by evaporating at −20 °C in crystallization dishes. Light blue needle-like crystals were generated and quickly turned metallic gold color at room temperature. The structure of the final crystals was characterized by X-ray single-crystal diffraction to be PIDA•**1** co-crystals. The crystallographic data of PIDA•**1** has been deposited in the Cambridge Crystallographic Data Center (CCDC) with the accession number 2111447.

**Synthesis of C18-PMH-PEG**. Polyethylene glycol (PEG)-grafted poly (maleic anhydride-alt-1-octadecene) (C18-PMH-PEG) was synthesized according to the previous report[63]. 40 mg of poly(maleic anhydride-alt-1-octadecene) (C18-PMH) and 572 mg of PEG-NH$_2$ were dissolved in 2 mL of dichloromethane with 200 μL triethylamine (TEA). After 72 h of stirring, the dichloromethane solvent was removed and the remaining solid was dissolved in water to form a transparent solution, which was dialyzed against distilled water for 3 days in a dialysis bag with a molecular weight cut-off of 7 kDa to remove unreacted PEG-NH$_2$. The final freeze-dried product was stored at −20 °C for future use. $^1$H NMR (400 MHz, CDCl$_3$) δ 0.87 (br, CH$_3$ of C18-PMH), 1.24 (br, CH$_2$ of C18 chains), 3.63(br, CH$_2$

of PEG). The effective PEGylation degree (based on available carboxyl groups) of the final product was found to be 16.5%.

**Preparation of PIDA suspensions**. 200 mg of PIDA•1 co-crystals and 100 mg of C18-PMH-PEG were mixed in water (40 mL). The mixture was sonicated in the ice bath for 4 h with the ultrasonic cell disruptor (650 W, 60%). The suspensions were dialyzed against deionized water for 2 days in a dialysis bag with a molecular weight cut-off of 7 kDa to remove **1**. The dark blue suspensions were lyophilized and re-suspended to corresponding concentrations before every use.

**Preparation of PIDA fiducial marker**. 200 mg of PIDA co-crystals was added in 40 mL of ethanol and placed in the shaker for 30 min. The mixture was centrifuged at $1790 \times g$ for 15 min to remove **1**, which was soluble in ethanol. The remaining solid was washed with ethanol three times. Drying the solid in vacuum at room temperature yielded bulk PIDA. 3 mg of the bulk PIDA sample was cast into a cylindrical shape with a diameter of 1 mm and a length of 5 mm. The cylindrical PIDA was used as fiducial marker after sterilization.

**Cell toxicity assay**. The cytotoxicity of PIDA was assessed by MTT (Thiazolyl Blue Tetrazolium Bromide) assays. 4T1 cells, NIH 3T3 cells, and HEK 293 T cells (kindly provided by Chenhui Wang, Huazhong University of Science and Technology) were seeded in 96-well plates and cultured in standard medium containing 10% FBS and 1% antibiotics (penicillin, $10000$ U mL$^{-1}$, streptomycin 10 mg mL$^{-1}$) for 24 h (37 °C, 5% CO$_2$). The cells were then incubated with PIDA suspensions at various concentrations for 24 h. The supernatants were discarded and added with fresh standard medium, and then 10 μL of freshly prepared MTT (5 mg mL$^{-1}$) solution was added into each well. The MTT solution was carefully removed after 4 h of incubation, and DMSO (150 μL) was added into each well to dissolve all the formazan formed. The absorbance of MTT at 570 nm was measured. Cell viability was expressed by the ratio of the absorbance of the cells incubated with PIDA to those incubated with a culture standard medium.

**Hemolysis assay**. To 150 μL of 10% w/v red blood cells (RBCs) suspension, a PIDA suspension at specific concentration (150 μL) was added, followed by incubation at 37 °C for 3 h[64]. The mixture was centrifuged ($640 \times g$, 10 min) and the supernatant was transferred to a 96-well plate (150 μL per well). Hemoglobin release was measured by the microplate reader as the absorbance (A) at 540 nm. Triton X-100 and phosphate-buffered saline (PBS) were used as positive and negative controls, respectively. The hemolysis percentage was calculated as $(A_{sample} - A_{PBS})/(A_{triton} - A_{PBS}) \times 100\%$.

**Animals and orthotopic xenograft rat models**. All the animal experiments were approved by the Institutional Animal Care and Use Committee of Huazhong University of Science and Technology ([2021] IACUC Number: 2622) in this work. SD male rats (6–8-week old, 160–180 g) and the male Beagle (1 year old, 8 Kg) were acquired from the Laboratory Animal Centre of Hubei, China. The orthotopic allograft rat models were prepared by injecting Walker 256 malignant ascites (kindly provided by Bin Xiong, Huazhong University of Science and Technology) at a site on the inner thigh. The maximal tumor size in the experiments did not exceed the limit (1500 mm$^3$) of IACUC.

**MR Imaging**. MR imaging examinations were performed with a 3-T MR imaging system (Ingenia 3.0 T CX, Philips Healthcare, Best, the Netherlands), using a 12-channel rat coil (Zhongzhi medcoil, Suzhou, China). Before imaging, the rats were anaesthetized with an intraperitoneal injection of pentobarbital (3% [w/v] at 0.2 mL/100 g b.wt). The rats were placed in a head-first prone position. The abdomen of each rat was fixed with a belt of adhesive tape to limit respiratory motion. T2 weighted images were used in the coronal plane with Parameters TR 1080 ms TE 66 ms.

**CT imaging**. All CT scans were performed on the multi-detector CT scanner. The tube voltage was fixed at 70 kV, the pitch was 0.6 mm, and the collimator widths were $64 \times 0.6$ mm to acquire CT images. CT images were reconstructed based on the raw data with a matrix size of $512 \times 512$ as the axial images (Slice thickness: 0.6 mm or 2 mm; Slice interval: 0.4 mm) in the transverse slice direction. Before CT tests, the rats were anesthetized with an intraperitoneal injection of pentobarbital (3% [w/v] at 0.2 mL/100 g). All CT images were transferred to a post-processing workstation (Sygno Via, Siemens Healthcare, Erlangen, Germany). Image analysis and evaluation were performed with the 3D multi-viewer mode. CT value was measured by drawing the region of interest (ROI) to take the average value. 3D images post-processing reconstruction was performed with the dynamic volume rendering technology.

**CT-guided sample injection**. CT-guided puncture technology was used to accurately place the material at the target position. The target position was determined by CT imaging, and the syringe or puncture needle was used to place the material at the target position under CT image guidance.

**X-ray imaging**. After the rats were implanted with PIDA or Au marker via puncture needles under the guidance of CT image, X-ray imaging was used to evaluate the detectability of PIDA or Au marker. An X-ray image system (Fluoroscope compact, Siemens Healthcare, Erlangen, Germany) was used to generate an X-ray image for all measurements acquired in this study. The rats were photographed in the posterior-anterior position or the lateral position. The tube voltage was fixed at 45.8 kV, and the tube current was set at 741 mA, 9 ms.

**Robotic stereotactic body radiation therapy**. To immobilize the position, a vacuum cushion was used for the Beagle. Thin-slice planning CT (Tube voltage: 80 kV; Slice thickness: 1 mm) served as a basis for the treatment planning. The treatment planning and dose calculations for RRT were performed with MultiPlan 5.x/Precision 2.0, while the dose was applied using the CyberKnife® Robotic Radiosurgery System (CyberKnife system 10.5.x, Accuray Inc., Sunnyvale, CA, USA). The Ray-Tracing algorithm for dose calculation was used routinely. The stereo X-ray system acquired images (Tube voltage: 90 kV; Tube current: 500 mA, 40 ms) were used to establish the synchrony tracking model by determining the position of fiducial marker. Uncertainty (%) and correlation error (mm) were used to evaluate the accuracy and efficiency of PIDA marker tracking. The value of the uncertainty (%) threshold parameter over 40% or correlation error over 5 mm was considered unacceptable according to the Cyberknife® system's operation manual[60].

**Phantom analysis and evaluation**. A CT Electron density and image quality phantom (Model 062 M, Computerized Imaging Reference Systems, Inc. Virginia, USA) was used to evaluate the CT artifacts and their effects on radiation dose calculation. The Au marker and PIDA were embedded as indicated. CT images of phantom were transferred to Halcyon™ Radiotherapy system (Varian Medical Systems, Inc. USA). The relative CT intensities were derived from corresponding CT images using ImageJ software. The radiation dose calculations were conducted in the same radiotherapy planning using the Halcyon™ Radiotherapy system. The methods are available from the corresponding author upon request.

**Reporting summary**. Further information on research design is available in the Nature Research Reporting Summary linked to this article.

## Data availability

The X-ray crystallographic coordinates for structures reported in this study have been deposited at the Cambridge Crystallographic Data Centre (CCDC), under a deposition number 2111447. These data can be obtained free of charge from The Cambridge Crystallographic Data Centre via www.ccdc.cam.ac.uk/data_request/cif. All relevant data are available within the article, Supplementary Information, or Source Data file. The imaging data are available from the corresponding author upon request.

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

## Acknowledgements

The authors thank Yuncheng Cai for the help on crystal structure analysis, and Ming Yang, Huageng Liang, and Junwei Tong for the help on animal experiments. We also thank Dr. Chenhui Wang and Dr. Bin Xiong for kindly providing experimental cell lines. This work is supported by the National Natural Science Foundation of China (21877042, L.L.), the National Basic Research Plan of China (2018YFA0208903, L.L.), Huazhong University Startup Fund (L.L.), and the Natural Science Foundation of Hubei Province of China (2021CFB442, X. L.). We thank the Analytical and Testing Center of Huazhong University of Science and Technology for related analysis.

## Author contributions

M.Y., X.L., and Z.L. contributed equally to this work. M.Y., X.L., and Z.L. designed and performed the experiments. Y.G., S.T., Z.L., and Y. W. provided helpful suggestions for this work. J.L. participated in some of the animal work. X.L., F.M. and L.L. conceived and obtained funding for the project, oversaw the research, and wrote the paper. All authors discussed the results and commented on the paper.

## Competing interests

The authors declare no competing interests.
