## [Peer Review File · Nature Communications]

REVIEWERS' COMMENTS

Reviewer #1 (Remarks to the Author):

I read carefully the manuscript of Mingming Yin et al. entitled "Precisely translating computed tomography diagnosis accuracy into therapeutic intervention by a carbon-iodine conjugated Polymer". This manuscript is of high scientific quality. Moreover, this manuscript is well written and the results are clearly presented and discussed.

Due to its unusually high iodine payload, the PIDA under the aggregation and nanofibrous poly(diiododiacetylene) can amplify computed tomography for over 10 folds with improved geometrical and positional stability in vivo. Due to its intrinsic properties, this polymer can excel clinical fiducial markers of imaging-guided radiotherapy in efficiency and exhibit in the studies led by the authors of this manuscript successful guidance of robotic radiotherapy on Beagles.

This manuscript should be published as soon as possible due to its excellent results.

Reviewer #2 (Remarks to the Author):

This manuscript described the development and evaluation of Carbon-Iodine 2 Conjugated Polymer that could be used to guide preoperative planning and visualization-guided surgery. Its application as fiducial markers of imaging-guided radiotherapy was also explored. Although various experiments have been well designed, most of the experiments stay at early stage proof of principle stage and the impacts would be very limited. It may represent technical improvement of current technology on preoperative planning and fiducial marker. There are a few concerns.

1. For preoperative planning and visualization-guided surgery, various other technologies are being developed. The advantage over gold or silver nanoparticles, quantum dots etc are not thoroughly explored or discussed. For example, gold nanoparticles are well known to be biocompatible which also induces high signal in CT.
2. Nanofiber structure are known to cause inflammation at the injected area. The high loading of I in the polymer likely would make it have poor solubility. Although major organ may not see apparent toxicity, it is important to check injected area.

Reviewer #3 (Remarks to the Author):

This manuscript describes a novel CT/radiographic contrast agent that demonstrated increased attenuation and longer local retention than water soluble iodine contrast, as well as a blue color visible to the naked eye. Potential significant applications include use as a surgical marker after CT-guided injection of a lesion, as well as use as a fiducial marker to increase accuracy of treatment delivery during radiation therapy. The methodology is sound, well-described and reproducible. The route of injection should be described earlier in the manuscript to clarify that this is a locally injected agent and not administered intravenously.

The study agent compared favorably with water-soluble iodine contrast in terms of attenuation and

local retention, however comparison with an oil-based contrast agent such as ethiodized oil may have been more appropriate as a water-soluble CT contrast agent would not be expected to remain locally after IM injection and would likely not be used as a surgical marker given that characteristic. I expect that the study agent would still compare favorably to a product such as Lipiodol given the reported lack of local tissue reaction and visibility to the naked eye.

The use as a fiducial marker that has the potential to bind to soft tissue is promising. As the authors point out, currently used fiducial markers may migrate. The conclusion that the study agent shows positional stability relative to liver tissue has some support from the results; histology of the liver tissue with the implanted PIDA would have provided additional support. Given the similarity of use in the Beagle to proposed use in human patients undergoing radiation therapy, serial blood cell and biochemical blood testing on the dog would have been useful to report.

The results of this original study are relevant to the fields of human and veterinary radiation oncology.

Reviewer #4 (Remarks to the Author):

In this work, the authors report the synthesis of poly(diiododiacetylene) (PIDA), a conjugated polymer made of iodine and carbon, which, however, is not new. On the other hand, the authors, report the use of PIDA for ultraefficient X-ray computed tomography (CT) and imaging-guided radiotherapy. They demonstrate that PIDA outperforms similar commercially available contrast agents. In vitro and in vivo studies are carried out very well. I would like to recommend publication of this work in Nature after the minor changes proposed below:

1. Page 4, line 76: I would remove host-guest because this terminology is more suitable to systems that are encapsulated, where the host encapsulates the guest. I understand that this was taken from the literature, but it is wrong. What the authors pursued is co-crystal strategy where the two components self-assembled in halogen-bonded 1D chains. Please remove host-guest from the manuscript.
2. There are a few misspellings throughout the manuscript that should be corrected, e.g., page 2, line 40, availability.
3. Page 5, lines 95-96, the definition of halogen bonding is wrong, please refer to the IUPAC accepted definition (see PAC 2013, 1711 and Chem. Rev. 2016, 2478). Also, more recent references on halogen bonding need to be added, especially those involving photoreactions and iodoalkyne derivatives.

Response to Reviewer 1

I read carefully the manuscript of Mingming Yin et al. entitled "Precisely translating computed tomography diagnosis accuracy into therapeutic intervention by a carbon-iodine conjugated Polymer". This manuscript is of high scientific quality. Moreover, this manuscript is well written and the results are clearly presented and discussed.

Due to its unusually high iodine payload, the PIDA under the aggregation and nanofibrous poly(diiododiacetylene) can amplify computed tomography for over 10 folds with improved geometrical and positional stability in vivo. Due to its intrinsic properties, this polymer can excel clinical fiducial markers of imaging-guided radiotherapy in efficiency and exhibit in the studies led by the authors of this manuscript successful guidance of robotic radiotherapy on Beagles.

This manuscript should be published as soon as possible due to its excellent results.

Response: We are very grateful to the reviewer for the kind encouragement and comments on our work. Such recognition has provided us great motivation in the future follow-up research.

Response to Reviewer 2

This manuscript described the development and evaluation of Carbon-Iodine 2 Conjugated Polymer that could be used to guide preoperative planning and visualization-guided surgery. Its application as fiducial markers of imaging-guided radiotherapy was also explored. Although various experiments have been well designed, most of the experiments stay at early stage proof of principle stage and the impacts would be very limited. It may represent technical improvement of current technology on preoperative planning and fiducial marker. There are a few concerns.

1. For preoperative planning and visualization-guided surgery, various other technologies are being developed. The advantage over gold or silver nanoparticles, quantum dots etc are not thoroughly explored or discussed. For example, gold nanoparticles are well known to be biocompatible which also induces high signal in CT.

Response: We really thank the reviewer for having a high opinion on the experiment design in this manuscript. We also appreciate the valuable comments on the manuscript. We agree that many other technologies have been developed for preoperative planning and visualization-guided surgery. These strategies, including metal nanoparticles and quantum dots, are also important in some cases. There are three major advantages of PIDA over these methods. First, PIDA is an organic polymer with good biocompatibility. Although the reviewer pointed out that gold nanoparticles are also biocompatible, most current clinically available CT contrast agents are

iodine-based organic compounds or polymers. The clinical safety of metal nanoparticles or quantum dots still needs to be validated thoroughly. Second, PIDA has a deep blue color because of its highly planar conjugated backbone, so that it is easily distinguishable from the surrounding tissues by naked eyes. This notable dual visibility by both CT imaging and naked eyes warranted the accurate translation of diagnostic findings into surgical guidance. On the contrary, nanoparticles or quantum dots are difficult to achieve this dual visibility. Finally, PIDA can be used as a fiducial marker in imaging-guided radiotherapy (IGRT), with excellent biocompatibility, fixed positioning, and minimal artifacts while maintaining an effectively high CT intensity. These characteristics are critical for the precise calculation of dose distribution in treatment planning systems, as well as for the accurate radiation delivery in robotic IGRT. In comparison, Au-based fiducial markers often lead to severe side effects, including inflammation, swollen, bleeding, and consequently position shift. Their permanent *in vivo* stay is also a threat in prognosis and management. In addition, Au-based fiducial markers generate streak artifacts in CT images, therefore hindering the visibility of surrounding tissues and affecting the accuracy of dose distribution calculation and treatment planning.

2. Nanofiber structure are known to cause inflammation at the injected area. The high loading of I in the polymer likely would make it have poor solubility. Although major organ may not see apparent toxicity, it is important to check injected area.

Response: We highly appreciate this valuable suggestion by the reviewer. We agree that nanofiber structures may cause inflammation at the injection site. To minimize the possible toxicity due to the poor solubility of PIDA, we have prepared the formulation in which PIDA is well suspended. In addition, as the reviewer suggested, we have checked the inject area to further verify the biocompatibility of PIDA nanofibers, as shown in Fig. R1 (Supplementary Fig. 16 in the Supplementary Information). The H&E and TUNEL staining images of the tissues at the injection site revealed that there was some damage in the tissue, which was caused by the injection needle. However, the tissues at the injection site recovered significantly on Day 7 post injection, and completely healed after 1 month. No PIDA-caused toxicity was observed. We have added this discussion in the revised manuscript on Page 21 (highlighted in red) as well as in the Supplementary Information (Supplementary Fig. 16).

Fig. R1 H&E and TUNEL staining images of the tissues at the injection site of PIDA. Scale bars: 100 μ m. The experiment was carried out on biologically independent samples.

Responses to Reviewer 3

This manuscript describes a novel CT/radiographic contrast agent that demonstrated increased attenuation and longer local retention than water soluble iodine contrast, as well as a blue color visible to the naked eye. Potential significant applications include use as a surgical marker after CT-guided injection of a lesion, as well as use as a fiducial marker to increase accuracy of treatment delivery during radiation therapy. The methodology is sound, well-described and reproducible.

The route of injection should be described earlier in the manuscript to clarify that this is a locally injected agent and not administered intravenously.

Response 1: We highly appreciate the reviewer for the elaborate review and valuable comments. We have added the description of local injection of PIDA in the Introduction (Page 4, highlighted in red).

The study agent compared favorably with water-soluble iodine contrast in terms of attenuation and local retention, however comparison with an oil-based contrast agent such as ethiodized oil may have been more appropriate as a water-soluble CT contrast agent would not be expected to remain locally after IM injection and would likely not be used as a surgical marker given that characteristic. I expect that the study agent would still compare favorably to a product such as Lipiodol given the reported lack of local tissue reaction and visibility to the naked eye.

Response: We really appreciate the reviewer for this helpful suggestion. PIDA is insoluble in common solvents, and was dispersed in aqueous media for the in vivo use, so that we chose the aqueous solution of iohexol as a negative control, which can better illustrate the retention of PIDA after local injection. For oil-based Lipiodol, as the reviewer pointed out, it is not visible to the naked eyes in surgery as PIDA. In addition, there were literatures reporting its in vivo movements. For example, Lipiodol was reported to move with bladder expansion and contraction when used as a fiducial marker for IGRT (*Int. Braz. J. Urol.* **2014**, 40, 190-197).

The use as a fiducial marker that has the potential to bind to soft tissue is promising. As the authors point out, currently used fiducial markers may migrate. The conclusion that the study agent shows positional stability relative to liver tissue has some support from the results; histology of the liver tissue with the implanted PIDA would have provided additional support. Given the similarity of use in the Beagle to proposed use in human patients undergoing radiation therapy, serial blood cell and biochemical blood testing on the dog would have been useful to report. The results of this original study are relevant to the fields of human and veterinary radiation oncology.

Response: Thank the reviewer for the valuable suggestions. The biosafety of PIDA has been well verified in rats. We totally agree that biochemical blood tests of the dog would be useful. Since we are planning clinical translational studies of PIDA, we will provide serial blood cell and biochemical blood testing on the dog or even on human in our following studies.

Responses to reviewer 4

In this work, the authors report the synthesis of poly(diiododiacetylene) (PIDA), a conjugated polymer made of iodine and carbon, which, however, is not new. On the other hand, the authors, report the use of PIDA for ultraefficient X-ray computed tomography (CT) and imaging-guided radiotherapy. They demonstrate that PIDA outperforms similar commercially available contrast

agents. In vitro and in vivo studies are carried out very well. I would like to recommend publication of this work in Nature after the minor changes proposed below:

Response: We are very grateful to the reviewer for their careful reading and pointing out the novelty of our work. We also highly appreciate the reviewer's suggestions for strengthening our work.

1. Page 4, line 76: I would remove host-guest because this terminology is more suitable to systems that are encapsulated, where the host encapsulates the guest. I understand that this was taken from the literature, but it is wrong. What the authors pursued is co-crystal strategy where the two components self-assembled in halogen-bonded 1D chains. Please remove host-guest from the manuscript.

Response: Thank the reviewer for the comments. As the reviewer suggested, we have removed all "host-guest" descriptions in the manuscript.

2. There are a few misspellings throughout the manuscript that should be corrected, e.g., page 2, line 40, availability.

Response 2: Thank the reviewer for pointing out this mistake. We apologize for these errors and corrected it accordingly. The entire manuscript has been checked more carefully to avoid similar mistakes.

3. Page 5, lines 95-96, the definition of halogen bonding is wrong, please refer to the IUPAC accepted definition (see PAC 2013, 1711 and Chem. Rev. 2016, 2478). Also, more recent references on halogen bonding need to be added, especially those involving photoreactions and iodoalkyne derivatives.

Response: Thank the reviewer for the helpful comments. We did understand the definition of halogen bonding, which refers to the noncovalent interaction between halogen atoms (Lewis acids) and neutral or anionic Lewis bases (Acc. Chem. Res 2005, 38, 386-395). We have modified the statement about halogen bonding in the revised manuscript to make it more accurate (Page 5, highlighted in red). In addition, we have cited more recent references on halogen bonding in the revised manuscript (e.g., PAC 2013, 1711; Chem. Rev. 2016, 2478)